# Molecular Signature of Small Cell Lung Cancer after Treatment Failure: The *MCM* Complex as Therapeutic Target

**DOI:** 10.3390/cancers13061187

**Published:** 2021-03-10

**Authors:** Shunsuke Misono, Keiko Mizuno, Takayuki Suetsugu, Kengo Tanigawa, Nijiro Nohata, Akifumi Uchida, Hiroki Sanada, Reona Okada, Shogo Moriya, Hiromasa Inoue, Naohiko Seki

**Affiliations:** 1Department of Pulmonary Medicine, Graduate School of Medical and Dental Sciences, Kagoshima University, Kagoshima 890-8520, Japan; k8574402@kadai.jp (S.M.); keim@m.kufm.kagoshima-u.ac.jp (K.M.); taka3741@m2.kufm.kagoshima-u.ac.jp (T.S.); k8802984@kadai.jp (K.T.); akiuchi@m3.kufm.kagoshima-u.ac.jp (A.U.); k8173956@kadai.jp (H.S.); inoue@m2.kufm.kagoshima-u.ac.jp (H.I.); 2Merck Sharp and Dohme (MSD) K.K., Tokyo 102-8667, Japan; nijiro.nohata@merck.com; 3Department of Functional Genomics, Chiba University Graduate School of Medicine, Chuo-ku, Chiba 260-8670, Japan; reonaokada@chiba-u.jp; 4Department of Biochemistry and Genetics, Chiba University Graduate School of Medicine, Chuo-ku, Chiba 260-8670, Japan; moriya.shogo@chiba-u.jp

**Keywords:** small cell lung cancer, cell cycle pathway, *MCM2*, *MCM4*, *MCM6*, *MCM7*

## Abstract

**Simple Summary:**

Small cell lung cancer (SCLC) is a fatal malignant tumor with a poor prognosis for patients who relapse after first-line treatment. There are few effective treatments for SCLC patients with relapse. Elucidation of the molecular network related to treatment resistance is an important issue for this disease. In this study, the molecular signature of SCLC specimens after treatment failure was generated. Several pathways, e.g., “cell cycle”, “homologous recombination”, “DNA replication”, and “p53 signaling” were identified as the enriched pathways in this signature. Aberrant expression of MCM2, MCM4, MCM6, and MCM7 were detected in SCLC clinical specimens after treatment failure. This signature contains molecules involved in treatment resistance and will contribute to the study of SCLC molecular pathogenesis.

**Abstract:**

Small cell lung cancer (SCLC) is a highly aggressive cancer, and patients who become refractory to first-line treatment have a poor prognosis. The development of effective treatment regimens is urgently needed. In this study, we identified a gene expression signature of SCLC after treatment failure using SCLC clinical specimens (GEO accession number: GSE162102). A total of 1,136 genes were significantly upregulated in SCLC tissues. These upregulated genes were subjected to KEGG pathway analysis, and “cell cycle”, “Fanconi anemia”, “alcoholism”, “systemic lupus erythematosus”, “oocyte meiosis”, “homologous recombination”, “DNA replication”, and “p53 signaling” were identified as the enriched pathways among the genes. We focused on the cell cycle pathway and investigated the clinical significance of four genes associated with this pathway: minichromosome maintenance (*MCM*) 2, *MCM4*, *MCM6*, and *MCM7*. The overexpression of these *MCM* genes was confirmed in SCLC clinical specimens. Knockdown assays using siRNAs targeting each of these four *MCM* genes showed significant attenuation of cancer cell proliferation. Moreover, siRNA-mediated knockdown of each *MCM* gene enhanced the cisplatin sensitivity of SCLC cells. Our SCLC molecular signature based on SCLC clinical specimens after treatment failure will provide useful information to identify novel molecular targets for this disease.

## 1. Introduction

Lung cancer is the one of the most malignant cancers, with high morbidity and mortality rates (approximately 2,100,000 people were diagnosed, and 1,800,000 patients died worldwide in 2018) [1]. Lung cancer is divided into two subtypes based on histological classification: small cell lung cancer (SCLC) and non-small lung cell cancer [2]. Approximately 15% of lung cancer patients are diagnosed with SCLC, and due to rapid progression and strong invasiveness of SCLC tumors, approximately 70% of patients with SCLC have metastasis (extensive disease (ED)-SCLC) by the time of diagnosis [3].

Platinum-based chemotherapy is used as the initial treatment for patients with ED-SCLC, and it has improved survival [4]. A clinical feature of SCLC is that it responds efficiently to treatment initially but then acquire drug resistance during treatment [5,6]. There are few effective treatments for ED-SCLC patients with relapse, who rarely survive beyond 2 years (range 7–10 months) [7]. Clarifying the molecular mechanism by which cancer cells acquire drug resistance is an important issue for this disease.

SCLC is rarely treated surgically and obtaining tumor samples suitable for genome-based studies is especially difficult for ED-SCLC. Regardless, several genome-based studies have been conducted so far to elucidate the malignant phenotype of SCLC cells [8,9]. In recent studies, expression data deposited in the Gene Expression Omnibus (GEO) database were downloaded and reanalysed for differentially expressed genes and pathways in SCLC tissues [10,11,12]. For example, one of those studies merged four GEO datasets (GSE60052, GSE43346, GSE15240, and GSE6044) and identified 20 SCLC-related genes (*S1PR1*, *MAD2L1*, *CDKN2A*, *STIL*, *NDC80*, *NCAPG*, *PAD51AP1*, *TTK*, *PRM2*, *EZH2*, *PRC1*, *UBE2C*, *RFC4*, *CENPF*, *TCP2A*, *HMGB3*, *TYMS*, *SOX4*, *MCM2,* and *SMC4*) [10].

Several previous studies have performed whole genome analyses using SCLC clinical specimens [13,14,15,16,17,18,19]. It has been reported that SCLC has high mutation rates and genomic instability cause to tobacco carcinogens [15,20]. The tumor-suppressor genes *TP53* and *RB1* are the most frequently mutated genes in SCLC [13,14,15,16,18]. A meta-analysis of four studies reporting high-throughput sequencing data for SCLC revealed that *RB1* plays a pivotal role in tumor progression and the subsequent mutation burden of SCLC [19]. Those high-throughput sequencing studies revealed that several mutated genes are strongly involved in SCLC carcinogenesis (e.g., *SOX2*, *FGFR1*, *LRP1B*, *KIAA1211*, and *PTEN*) [13,14,16,17]. Moreover, mutations have been identified in several genes involved in the *PI3K*/*AKT*/*mTOR* pathway (e.g., *PIK3CA*, *PTEN*, *AKT*, *RICTOR*, and *mTOR*) in SCLC [15]. A large-scale cohort study of SCLC revealed that the presence of genetic alterations in the *PI3K*/*AKT*/*mTOR* pathway are strongly associated with poor survival of patients with ED-SCLC [18]. Those studies suggested that activation of *PI3K*/*AKT*/*mTOR*-mediated signaling enhances SCLC tumorigenesis, and that the involved genes are promising therapeutic targets for SCLC.

In this study, we identified a SCLC expression signature using clinical specimens obtained from SCLC patients after treatment failure. Normal tissues, primary lesions, and metastatic lesions were collected from three patients (autopsy specimens), and comprehensive gene expression analyses were performed. Our mRNA expression signature data was deposited in the GEO (Gene Expression Omnibus: GSE 162102) database and is accessible.

To discover genes involved in the aggressive nature and drug resistance of SCLC cells, we focused on genes that are highly expressed in cancer cells. The highly expressed genes were found to be enriched in “cell cycle”, “Fanconi anemia”, “alcoholism”, “systemic lupus erythematosus”, “oocyte meiosis”, “homologous recombination”, “DNA replication”, and “p53 signaling” pathways. Next, we performed functional analyses focusing on four minichromosome maintenance (*MCM*) genes (*MCM2*, *MCM4*, *MCM6*, and *MCM7*), which are involved in the cell cycle and cell replication. The replisome is a complex molecular machinery that are essential for DNA replication and are therapeutic target for human cancers. The MCM-family are major component proteins of DNA replisome [21]. The aberrant expression of these *MCM* genes contributed to an aggressive nature in SCLC cells. Furthermore, siRNA-mediated knockdown of *MCM* genes enhanced the cisplatin sensitivity of SCLC cells. There have been few reports of gene expression profile in treatment failure of SCLC clinical specimens. This signature contains molecules involved in treatment resistance and will contribute to the study of SCLC molecular pathogenesis.

## 2. Materials and Methods

### 2.1. Collection of SCLC Autopsy Specimens and Cell Lines

In this study, all SCLC and normal lung tissue specimens were obtained from three patients who died of drug-resistant SCLC with metastatic lesions at Sendai Medical Association Hospital in 2018. SCLC cell lines H82 and SBC-3 were purchased from the American Type Culture Collection (Manassas, VA, USA) and the Japanese Cancer Research Resources Bank (Osaka, Japan), respectively.

### 2.2. Immunohistochemistry and Western Blot Analysis

Immunostaining was performed with a VECTASTAIN^®^ Universal Elite ABC Kit (catalog no.: PK-6200; Vector Laboratories, Burlingame, CA, USA) using SCLC tissues obtained by autopsy. The expression of synaptophysin, chromogranin A and MCM family members (MCM2, MCM4, MCM6 and MCM7) in normal lung tissue was obtained from the Human Protein Atlas (version 20.0; http://www.proteinatlas.org/ (accessed on 5 February 2021)). In western blotting, cell lysates were prepared using a RIPA buffer (catalog no.: sc-24948, Santa Cruz Biotechnology Inc., Dallas, TX, USA). A total of 20 µg protein was separated on SuperSep ^TM^Ace (7.5%, 13 well) (FUJIFILM Wako Pure Chemical Corporation, Osaka, Japan) and transferred onto polyvinylidene fluoride membranes (catalog no.: PPVH00010, Merck KGaA, Darmstadt, Germany). The procedures have been described previously [22,23,24,25]. The primary antibodies used are shown in Appendix A.

### 2.3. Identification of the mRNA Expression Signature for Treatment Failure of SCLC

In this study, we extracted total RNA from SCLC autopsy specimens and evaluated gene expression using Agilent SurePrint G3 Human GE v3 8x60K microarrays. The raw microarray data were registered in GEO (GSE 162102). Our selection strategy is presented in Appendix A.

### 2.4. Transfection of siRNAs into SCLC Cells and Functional Assays

The cell proliferation assay was determined by XTT assays using a Cell Proliferation Kit (Biological Industries, Beit-Haemek, Israel). In the cell cycle assay, SCLC cells were treated with BD Cycletest^TM^ Plus DNA Reagent Kit (BD Biosciences, Franklin Lakes, NJ, USA) according to the manufacturer’s protocol. Apoptotic cells were detected using a PE Active Caspase-3 Apoptosis Kit (BD Biosciences). These SCLC cells were analyzed using a flow cytometer (BD FACSCelesta^TM^ Flow Cytometer, BD Biosciences). BD FACSDiva Software (version 8.0.1.1, BD Biosciences) was used to examine the flow cytometry data. The procedures used for transfecting siRNAs and for the functional assays (cell proliferation, cell cycle and apoptosis) were described in our previous studies [22,23,24,25]. The reagents used are listed in Appendix A.

### 2.5. RNA Preparation and Quantitative Reverse-Transcription Polymerase Chain Reaction (qRT-PCR)

The methods used for RNA extraction from clinical specimens and cell lines and for qRT-PCR have been described previously [22,23,24,25]. In brief, the isolation of total RNA from clinical specimens was performed using TRI reagent (Cosmo Bio Co., Ltd., Tokyo, Japan) and total RNA of cell lines was extracted by Isogen II (NIPPON GENE Co., Ltd., Tokyo, Japan) according to the manufacturer’s instructions. The quantification and quality of total RNA were checked by NanoDrop 2000c spectrophotometer (Thermo Fisher Scientific Inc., Waltham, MA, USA). cDNA was synthesized using PrimeScript^TM^ RT Master Mix (catalog no.: RR036A, Takara Bio Inc., Shiga, Japan). Subsequently, we evaluated the expression of the gene by TaqMan Real-Time PCR Assays. qRT-PCR reactions were run using the StepOnePlus Real-Time PCR System (Applied Biosystems, Foster City, CA, USA). TaqMan probes and primers are shown in Appendix A.

### 2.6. Combined Treatment Effect on Cell Viability

To determine the effect of cisplatin combined with *MCM2*, *MCM4*, *MCM6*, or *MCM7* knockdown on cisplatin sensitivity, SCLC cell lines were seeded at 4000/well in a 96-well plate. The following day, cisplatin (0.001, 0.01, 0.1, 0.5, 1, 2, 5, 10, 25, or 100 µM) was administered for 72 h with or without transfection of the *MCM*-targeting siRNAs. Cell viability was assessed by XTT assay, and the IC_50_ of cisplatin was determined.

### 2.7. Statistical Analyses

Statistical analyses were performed using GraphPad Prism 7 (GraphPad Software, La Jolla, CA, USA) and JMP Pro 14 (SAS Institute Inc., Cary, NC, USA). One-way analysis of variance (ANOVA) and Tukey’s post-hoc test were used for multiple group comparisons.

## 3. Results

### 3.1. Clinical Course of SCLC Patients and Immunostaining of Autopsy Tissues

In this study, we obtained autopsy specimens from three patients who died of drug-resistant metastatic SCLC. The characteristics of the patients and their clinical specimens are shown in Table 1 and Table 2. The clinical courses of the patients are presented in Figure 1A–C.

To confirm the histological type, the autopsy specimens were stained with hematoxylin and eosin and subjected to immunostaining of synaptophysin, chromogranin A, and CD56, which are specific markers of neuroendocrine tumors including SCLC (Figure 2).

### 3.2. mRNA Expression Signature of SCLC after Treatment Failure

Using Agilent SurePrint G3 Human GE v3 8 × 60 K microarrays, the gene expression signature of SCLC after treatment failure was determined. Among the genes differentially expressed between the cancer and normal lung tissues, a total of 1136 genes upregulated in the cancer tissues (Log fold change > 2) were extracted. Subsequently, KEGG pathway analysis was performed using the GeneCodis program (https://genecodis.genyo.es/ (accessed on 4 December 2020)), revealing that these genes are associated with the following pathways, “cell cycle”, “Fanconi anemia”, “alcoholism”, “systemic lupus erythematosus”, “oocyte meiosis”, “homologous recombination”, “DNA replication”, and “p53 signaling” (Table 3 and Appendix A). The microarray data were deposited in the GEO database (accession number: GSE162102).

### 3.3. Expression of MCM2, MCM4, MCM6, MCM7 in SCLC Tissues

Of the upregulated genes identified in SCLC tissues, we focused on *MCM* family genes (*MCM2*, *MCM4*, *MCM6*, and *MCM7*), which are involved in the cell cycle pathway (Table 4).

We confirmed by immunohistochemistry that the expression of MCM2, MCM4, MCM6, and MCM7 was significantly upregulated in SCLC specimens compared with normal tissues (case no. 1–3) (Figure 3 and Appendix A).

### 3.4. Effect of MCM2, MCM4, MCM6 and MCM7 Knockdown on SCLC Cells

To identify the functions of *MCM2*, *MCM4*, *MCM6*, and *MCM7*, we performed siRNA-mediated knockdown assays in SBC-3 and H82 cells. RT-PCR and Western blotting showed that expression levels of both mRNA and protein were markedly reduced by both siRNAs (Appendix A). Cell proliferation assays showed the reduced growth of SCLC cells transfected with siRNAs targeting *MCM2*, *MCM4*, *MCM6*, or *MCM7* compared with those transfected with the control siRNA (Figure 4).

We also performed cell cycle and apoptosis assays. The cell cycle assay demonstrated an increased proportion of cells in the G0/G1 phase after knockdown using si-*MCM2*-1 and si-*MCM4*-1 and an increased proportion of cells in the G2/M phase after knockdown using si-*MCM6*-1 and si-*MCM7*-1 in both SCLC cell lines (Figure 5). Also, in SBC-3 cells, si-*MCM6*-2 and si-*MCM7*-2 transfected cells increased proportion of cells in the G2/M phase, while in H82 cells, si-*MCM6*-2 and si-*MCM7*-2 transfected cells increased proportion of cells in the G0/G1 phase.

In the apoptosis assay, *MCM4*, *MCM6*, or *MCM7* knockdown increased the percentage of apoptotic cells in both SCLC cell lines (Figure 6 and Appendix A).

### 3.5. Enhanced Cisplatin Sensitivity by Knockdown of MCM Family Members in SBC-3 Cells

To detect the effect of *MCM2*, *MCM4*, *MCM6*, or *MCM7* knockdown on cisplatin sensitivity, SBC-3 cells were treated with cisplatin only or cisplatin together with siRNAs targeting each *MCM* member. Cell viability was then assessed by XTT assay. All siRNAs significantly reduced the IC_50_ of cisplatin, suggesting increased sensitivity of SCLC cells to cisplatin after *MCM* knockdown (Figure 7).

## 4. Discussion

Due to the rapid progression and quite invasive nature, approximately 70% of SCLC patients are advanced stage at the time of initial diagnosis [3]. Platinum-based chemotherapy is the first-line treatment for patients with ED-SCLC [4,7]. SCLC cells respond to this treatment initially, but over the course of treatment, they acquire resistance to treatment, and most patients relapse. The prognosis of patients with relapse is less than 1 year [7]. In order to control cancer cells that have acquired treatment resistance, genome analysis research using specimens of patients who have become treatment resistant is indispensable. We have succeeded in creating a molecular signature using clinical specimens of SCLC patients with treatment failure. Although the number of cases is limited, it provides a valuable signature that has been rarely reported so far.

Large-scale analyses using clinical specimens have been scarce because many patients with SCLC are not indicated for surgery. Regardless, several studies have sought to elucidate the malignant features of SCLC cells. Based on the available gene expression data so far, bioinformatics analysis has revealed the genes and molecular pathways involved in SCLC malignant transformation [10,11,12]. Taking the previous reports together, several genes are highly expressed in cancer cells and integrally involved in SCLC pathogenesis, including *NDC80*, *BUB1B*, *KIF2C*, *CDC20*, *MAD2L1*, *TOP2A*, *PCNA*, *RFC4*, *CHEK1*, *TYMS*, *MCM2*, *CDKN3*, *MCM3*, *CDC6*, *KIF11*, *MSH2,* and *RAD21* [12].

Our present study based on KEGG pathway analysis showed that the genes upregulated in SCLC tissues were associated with the “cell cycle”, “Fanconi anemia”, “alcoholism”, “systemic lupus erythematosus”, “oocyte meiosis”, “homologous recombination”, “DNA replication”, and “p53 signaling” pathways. Notably, a previous study of differentially gene expression analysis of SCLC by using GEO datasets (GSE6044 and GSE11969) showed that DNA replication pathway including *MCM*-family was upregulated in SCLC patients [26]. These analyzes were SCLC specimens that have not been treated with anticancer drugs. These results strongly suggest that DNA replication pathway involved genes are therapeutic targets for SCLC. We focused on *MCM* family genes (*MCM2*, *MCM4*, *MCM6*, and *MCM7*), which were identified as having roles in the “cell cycle” and “DNA replication” pathways. Cancer cells exhibit dysregulation of the cell cycle, leading to uncontrollable cell proliferation. Therefore, aberrantly expressed genes involved in the cell cycle and DNA replication are potential therapeutic targets. The replisome is a complex molecular machinery that closely contributes to DNA replication. Notably, some of the genes that make up replisome were upregulated in treatment failure patients with SCLC, e.g., cell division cycle 7 (*CDC7*), cell division cycle 45 (*CDC45*), *MCM2*, *MCM4*, *MCM6*, and *MCM7*. Control of replisome is an important concept in cancer treatment. The MCM complex is a eukaryotic replicative helicase that functions as a molecular motor unwinding duplex DNA to generate single strands of DNA templates for replication [21]. The MCM family contains six proteins (MCM2–7), which form a heterohexameric complex [21]. Several kinases phosphorylate the MCM complex, leading to the recruitment of CDC45 and GINS family members and formation of replisomes, essential DNA replication units [27]. It has been reported that the MCM complex comprising MCM2–7 has various functions other than replication (e.g., transcription, replication checkpoint, and RNA splicing) depending on the phosphorylation sites and status [27,28,29,30].

In cancer cells, the aberrant expression of MCM family members has been reported in a wide range of cancers, and cancer cell malignant phenotypes can be suppressed by knockdown of these genes [31,32,33,34]. Our recent study showed that siRNA-mediated knockdown of *MCM4* attenuated the aggressiveness of lung adenocarcinoma cells [22]. In this study, siRNA-mediated knockdown of *MCM2*, *MCM4*, *MCM6*, and *MCM7* enhanced the cisplatin sensitivity of SCLC cells. Proteins of the replisome complex, including MCM proteins, are target molecules for SCLC treatment.

Ciprofloxacin is a broad-spectrum antibiotic that binds to DNA gyrases and exhibits antibacterial activity by inhibiting DNA replication. A previous study showed that ciprofloxacin at low concentrations blocked the DNA helicase activity of the MCM complex [35]. Based on high-throughput screening of 2 million compounds, AS4583 was found to inhibit the cell proliferation and DNA replication of cancer cells via the disruption of the MCM complex, especially *MCM2* [36]. Moreover, a structure–activity relationship study of AS4583 led to the discovery of RJ-LC-07-48, which is a more effective anticancer drug for treating lung cancer cells resistant to tyrosine kinase inhibitors [36]. Our expression signature revealed that cell division cycle 7 (*CDC7*) was overexpressed in SCLC tissues after treatment failure. CDC7 is a crucial kinase required for initiating the replication machinery [37], and its overexpression has been reported in various cancers [37,38,39]. Importantly, CDC7 phosphorylates the MCM complex to activate its helicase function [39,40,41]. In melanoma cells, a sustained CDC7 expression and associated MCM2–7 activation were observed in vemurafenib-resistant cells, and the CDC7 inhibitor TAK-931 was suggested as a therapeutic option for vemurafenib-resistant melanoma cells [39]. CDC7 overexpression and constitutive MCM2-7 activation might contribute to the molecular pathogenesis of drug resistance in SCLC cells. The development of drugs that control CDC7 and MCM-family activation may contribute to the treatment of SCLC patients.

In this study, we created a molecular signature of patients with SCLC who became treatment failure. This signature contains a number of molecules that may control treatment resistance. Based on this signature, the understanding of molecular pathogenesis for treatment-resistant SCLC will be accelerated.

## 5. Conclusions

In conclusion, we successfully identified an expression signature of SCLC after treatment failure using clinical specimens. Among the genes upregulated in SCLC tissues compared with normal tissues, “cell cycle” and “DNA replication” were identified as enriched pathways. DNA replisome-related genes, *MCM2*, *MCM4*, *MCM6*, *MCM7*, *CDC45*, and *CDC7*, were identified as upregulated in the SCLC tissues and may be potential therapeutic targets for treatment-resistant SCLC cells. The treatment-failure signature for SCLC derived from this study will contribute to elucidating the molecular pathogenesis of cancer cells that have acquired treatment resistance.

## Figures and Tables

**Figure 1 cancers-13-01187-f001:**
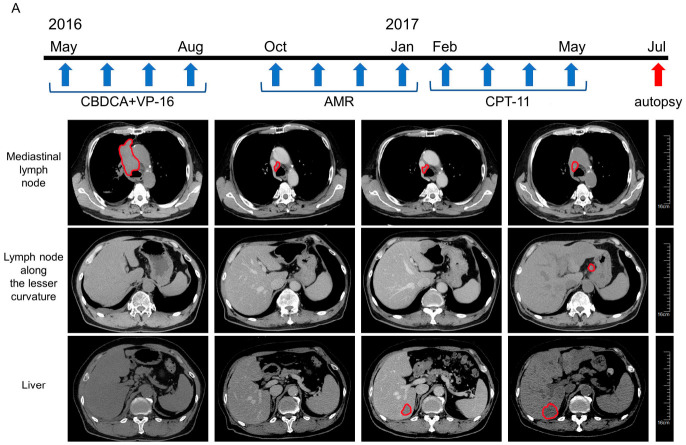
Clinical courses of the three patients with small cell lung cancer (SCLC). (**A**) The clinical course of case no. 1. A 66-year-old Japanese man visited due to facial edema and dyspnea in May 2016. Chest-abdomen computed tomography (CT) revealed a tumor located from the right hilum to the mediastinum, compression of the superior vena cava, and bone metastasis at the seventh lumbar vertebra. The patient was diagnosed with SCLC by sputum cytology. First-line chemotherapy with carboplatin (CBDCA) and etoposide (VP-16) was initiated, resulting in a partial response after four cycles. However, in October 2016, CT showed a new liver metastasis. Following four cycles of second-line chemotherapy with amrubicin (AMR), the patient underwent four cycles of third-line chemotherapy with irinotecan (CPT-11), but further disease progression occurred. In July 2017, the patient died, and autopsy was performed. We obtained tissue samples from the metastases in the mediastinal lymph node, lesser curvature lymph node, liver, and para-aortic lymph node. (**B**) The clinical course of case no. 2. A 77-year-old Japanese man presented to our hospital with fever in May 2016. CT identified a mass located from the right hilum to the lower lobe of the right lung. Bronchoscopic biopsy of the lung mass revealed SCLC, and fluorodeoxyglucose-positron emission tomography revealed multiple bone metastases. He underwent six cycles of first-line chemotherapy with CBDCA and VP-16 and experienced a partial response. At 6 months after discontinuation of chemotherapy, CT showed an increase in size of the tumor in the right lower lobe, as well as a malignant pleural effusion. Thus, second-line chemotherapy with CPT-11 was started, but further disease progression occurred. In August 2017, the patient died, and autopsy was performed. We obtained tissue samples from the tumors in the right lower lobe (primary lesion) and the right hilar lymph node metastasis. (**C**) Clinical course of case no. 3. A 65-year-old Japanese man had experienced coughing for 2 months before visiting our hospital in April 2017. CT revealed a mass in the upper lobe of the left lung, lymphadenopathy in the left hilum and mediastinum, and multiple nodules in the right lung. Bronchoscopy with transbronchial biopsy confirmed a diagnosis of SCLC. First-line chemotherapy with CBDCA and VP-16 was initiated, but the intrapulmonary metastases in the right upper lobe increased in size. A switch to second-line chemotherapy with paclitaxel had no effect. In August 2017, the patient died, and autopsy was performed. We obtained tissues from the tumors in the left upper lobe (primary lesion) and the right intrapulmonary metastases.

**Figure 2 cancers-13-01187-f002:**
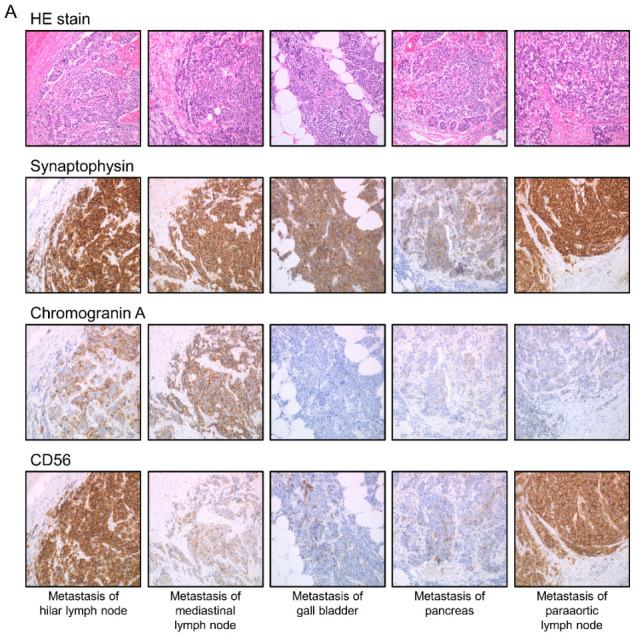
Histopathological findings in the three SCLC patients (**A**) case no. 1, (**B**) case no. 2, and (**C**) case no. 3. Micrographs were taken at 200× magnification.

**Figure 3 cancers-13-01187-f003:**
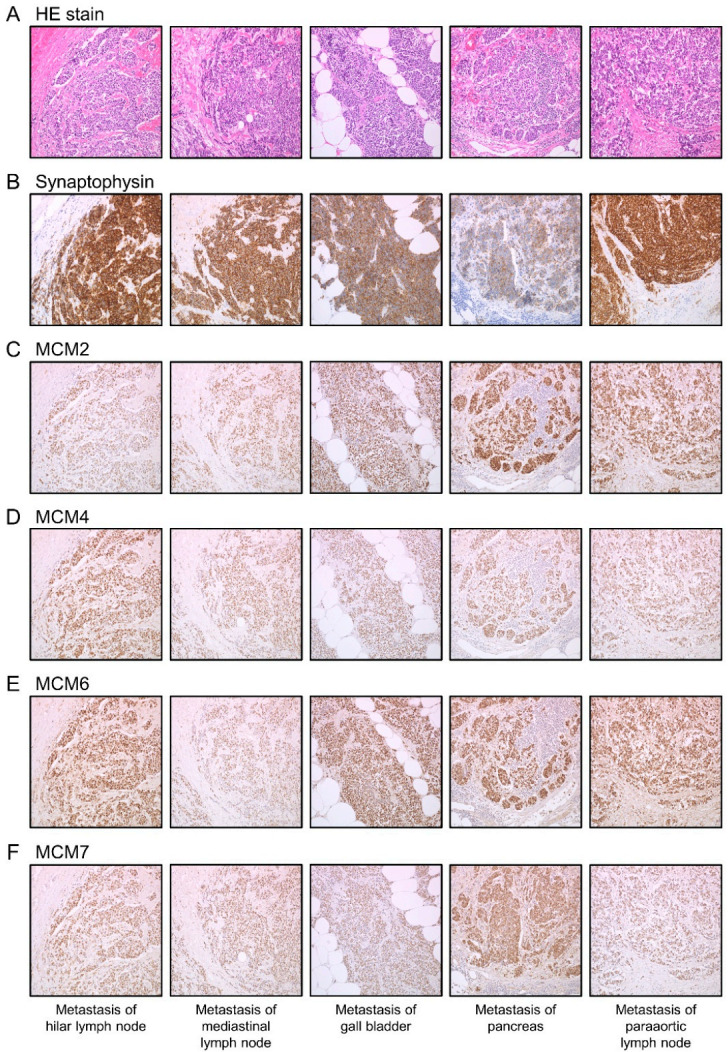
Histopathological findings and immunohistochemical analyses of MCM2, MCM4, MCM6, and MCM7 in case no. 1. Overexpression of (**C**) MCM2, (**D**) MCM4, (**E**) MCM6, and (**F**) MCM7 was observed in the nuclei of cancer cells at the same location as the tumor sites of (**A**) HE stain and the high expression sites of (**B**) Synaptophysin. Micrographs were taken at 200× magnification.

**Figure 4 cancers-13-01187-f004:**
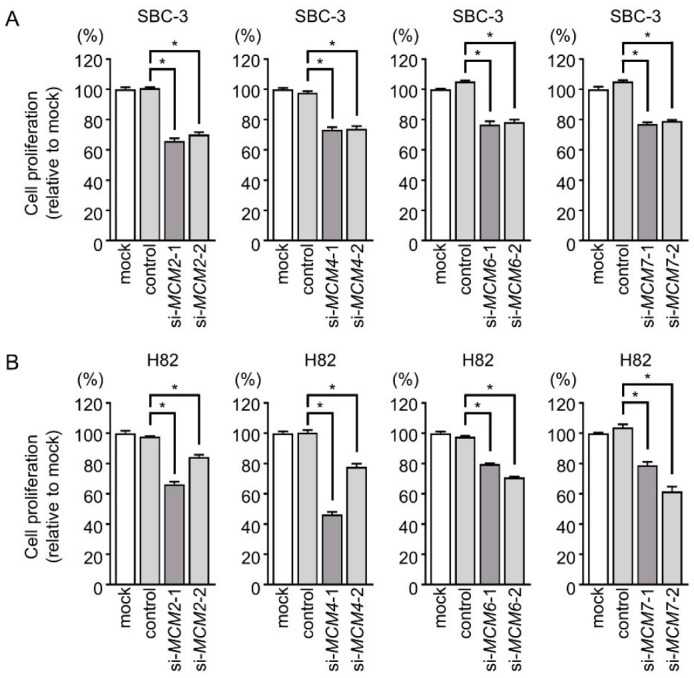
Effect of siRNA-mediated knockdown of *MCM* family members on the proliferation of SCLC cells. (**A**,**B**) Cell proliferation was evaluated by XTT assay. Cell proliferation was suppressed by transfection of siRNAs targeting *MCM2*, *MCM4*, *MCM6*, or *MCM7* in SBC-3 and H82 cells. *, *p* < 0.0001.

**Figure 5 cancers-13-01187-f005:**
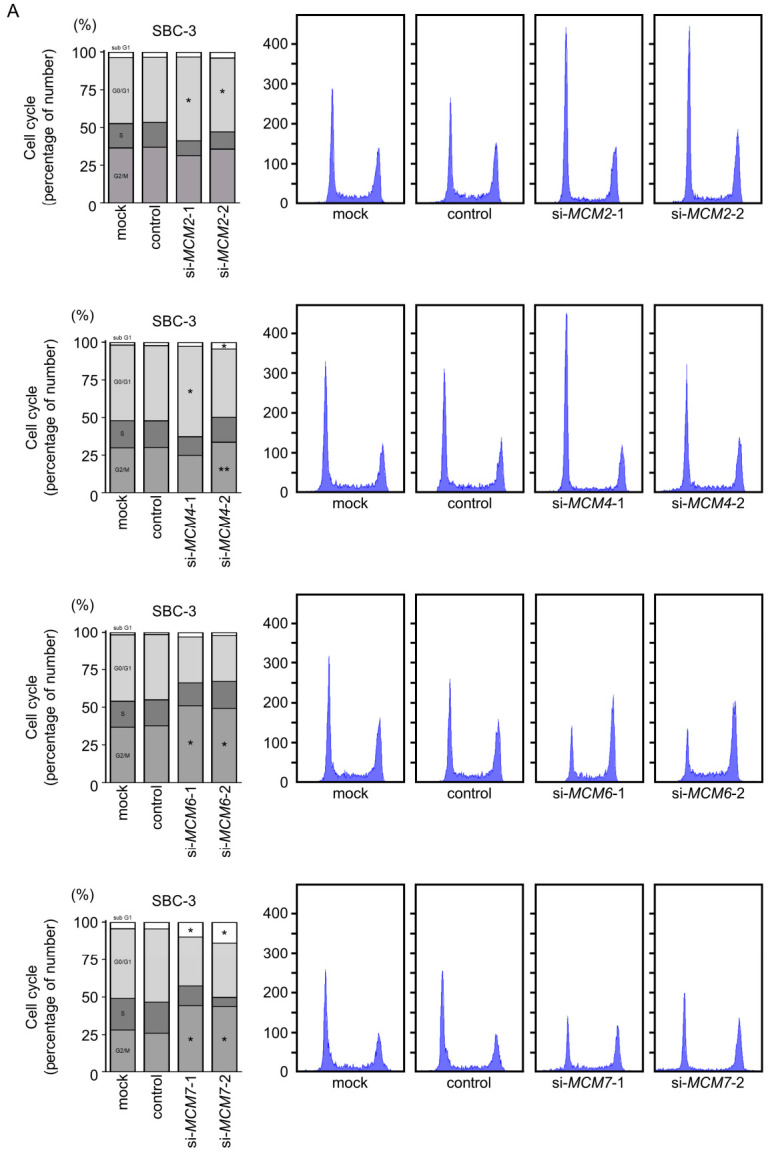
Effect of siRNA-mediated knockdown of *MCM* family members on cell cycle regulation in SCLC cells. (**A**) In SBC-3 cells, the cell cycle assay indicated G0/G1 phase arrest in cells transfected with siRNAs targeting *MCM2* or *MCM4* and G2/M phase arrest in cells transfected with siRNAs targeting *MCM6* and *MCM7*. (**B**) In H82 cells, si-*MCM2*-1 or si-*MCM4*-1 transfected cells increased proportion of cells in the G0/G1 phase and si-*MCM6*-1 or si-*MCM7*-1 transfected cells increased proportion of cells in the G2/M phase. Also, the number of cells in the G0/G1 phase were significantly elevated in si-*MCM6*-2 and si-*MCM7*-2 transfected cells compared with control cells. *, *p* < 0.0001, **, *p* < 0.01.

**Figure 6 cancers-13-01187-f006:**
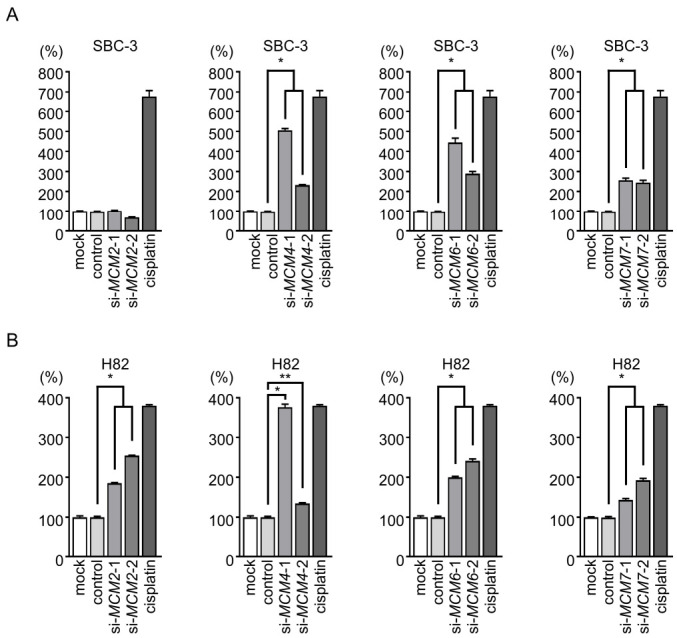
Effect of siRNA-mediated knockdown of *MCM* family members on apoptosis in SCLC cells. In both SBC-3 and H82 cells, the number of apoptotic cells was increased by transfection of siRNAs targeting *MCM4*, *MCM6*, or *MCM7*. Transfection of the *MCM2*-specific siRNAs caused apoptosis only in H82 cells. *, *p* < 0.0001, **, *p* < 0.01.

**Figure 7 cancers-13-01187-f007:**
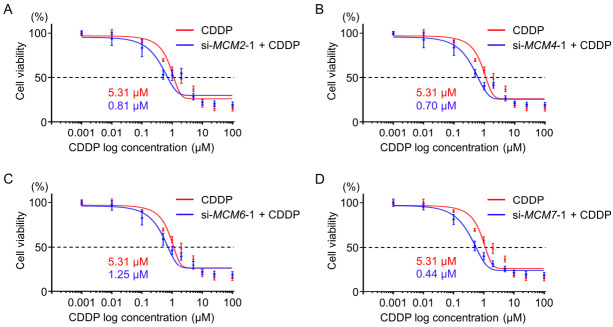
Effect of siRNA-mediated knockdown of *MCM* family members on cisplatin sensitivity. (**A**–**D**) The IC_50_ values of cisplatin in SBC-3 cells treated with cisplatin and siRNAs targeting each *MCM* gene for 72 h were determined by XTT assay. The knockdown of *MCM2*, *MCM4*, *MCM6* and *MCM7* increased the toxicity of cisplatin to SBC-3 cells.

**Table 1 cancers-13-01187-t001:** Clinical features of the SCLC patients.

No.	Sex	Age	BI	T	N	M	Stage	Therapy	Remarks
1	M	67	1440	4	2	1b	IV	Platinum based chemotherapy	Microarray expression analysis IHC staining
2	M	78	1180	2b	2	1b	IV	Platinum based chemotherapy	Microarray expression analysis IHC staining
3	M	65	990	2a	3	1a	IV	Platinum based chemotherapy	Microarray expression analysis IHC staining

IHC: immunohistochemistry, BI: Brinkman index.

**Table 2 cancers-13-01187-t002:** Site of sample collection for microarray expression analysis.

A. Cancer Tissues	*n* = 8	
Sample Name	Tissue	Case No.
SCLC-Pri-1	Primary lesion	2
SCLC-Pri-2	Primary lesion	3
SCLC-Meta-1	Lesser curvature lymph node metastasis	1
SCLC-Meta-2	Mediastinal lymph node metastasis	1
SCLC-Meta-3	Para-aortic lymph node metastasis	1
SCLC-Meta-4	Right hilar lymph node metastasis	2
SCLC-Meta-5	Liver metastasis	1
SCLC-Meta-6	Right intrapulmonary metastasis	3
**B. Normal lung tissues**	***n* = 4**	
**Sample name**	**Tissue**	**case no.**
SCLC N-1	Normal lung tissue	1
SCLC N-2	Normal lung tissue	1
SCLC N-3	Normal lung tissue	2
SCLC N-4	Normal lung tissue	3

**Table 3 cancers-13-01187-t003:** Significantly enriched annotations of the upregulated genes identified by microarray expression analysis of SCLC clinical specimens.

No. of Genes	*p*-Value	Annotations
28	2.28 × 10^−12^	(KEGG) 04110: Cell cycle
15	9.66 × 10^−8^	(KEGG) 03460: Fanconi anemia pathway
26	6.19 × 10^−7^	(KEGG) 05034: Alcoholism
21	2.12 × 10^−6^	(KEGG) 05322: Systemic lupus erythematosus
18	7.67 × 10^−5^	(KEGG) 04114: Oocyte meiosis
10	8.54 × 10^−5^	(KEGG) 03440: Homologous recombination
9	1.92 × 10^−4^	(KEGG) 03030: DNA replication
10	1.21 × 10^−2^	(KEGG) 04115: p53 signaling pathway

**Table 4 cancers-13-01187-t004:** Upregulated genes associated with the cell cycle pathway.

				Normalized Read Count (Log_2_)			
Entrez GeneID	Gene Symbol	Gene Name	Location	Log_2_ Fold Change	Normal Lung Tissues	SCLC Tissues	*p*-Value
7272	*TTK*	TTK protein kinase	6q14.1	3.67	−0.37	3.30	0.0034
9134	*CCNE2*	cyclin E2	8q22.1	3.57	0.05	3.62	0.0002
4173	*MCM4*	minichromosome maintenance complex component 4	8q11.21	3.33	0.20	3.54	0.0004
699	*BUB1*	budding uninhibited by benzimidazoles 1 homolog	2q13	3.18	−0.25	2.92	0.0040
701	*BUB1B*	budding uninhibited by benzimidazoles 1 homolog beta	15q15.1	3.08	−0.03	3.05	0.0018
9133	*CCNB2*	cyclin B2	15q22.2	3.05	0.05	3.10	0.0055
1029	*CDKN2A*	cyclin-dependent kinase inhibitor 2A	9p21.3	3.01	0.24	3.25	0.0047
4171	*MCM2*	minichromosome maintenance complex component 2	3q21.3	2.96	0.01	2.97	0.0008
995	*CDC25C*	cell division cycle 25 homolog C	5q31.2	2.87	−0.01	2.87	0.0045
891	*CCNB1*	cyclin B1	5q13.2	2.82	−0.06	2.76	0.0008
1869	*E2F1*	E2F transcription factor 1	20q11.22	2.82	−0.30	2.52	0.0100
9700	*ESPL1*	extra spindle pole bodies homolog 1	12q13.13	2.70	0.10	2.80	0.0007
8318	*CDC45*	cell division cycle 45 homolog	22q11.21	2.64	−0.29	2.34	0.0027
8317	*CDC7*	cell division cycle 7 homolog	1p22.2	2.62	0.01	2.63	0.0006
4085	*MAD2L1*	MAD2 mitotic arrest deficient-like 1	4q27	2.61	−0.19	2.42	0.0013
898	*CCNE1*	cyclin E1	19q12	2.61	0.06	2.67	0.0041
1111	*CHEK1*	CHK1 checkpoint homolog	11q24.2	2.57	0.00	2.57	0.0088
993	*CDC25A*	cell division cycle 25 homolog A	3p21.31	2.50	−0.15	2.34	0.0044
9232	*PTTG1*	pituitary tumor-transforming 1	5q33.3	2.40	0.07	2.47	0.0030
983	*CDK1*	cyclin-dependent kinase 1	10q21.2	2.40	0.05	2.45	0.0058
10926	*DBF4*	DBF4 homolog	7q21.12	2.22	0.06	2.28	0.0002
4176	*MCM7*	minichromosome maintenance complex component 7	7q22.1	2.20	0.13	2.33	0.0002
890	*CCNA2*	cyclin A2	4q27	2.17	0.03	2.20	0.0141
4175	*MCM6*	minichromosome maintenance complex component 6	2q21.3	2.05	−0.11	1.94	0.0002
6502	*SKP2*	S-phase kinase-associated protein 2 (p45)	5p13.2	2.04	0.23	2.27	0.0117
5111	*PCNA*	proliferating cell nuclear antigen	20p13	2.04	0.01	2.05	0.0017
5591	*PRKDC*	protein kinase, DNA-activated, catalytic polypeptide	8q11.21	2.01	−0.03	1.98	0.0071
10744	*PTTG2*	pituitary tumor-transforming 2	4p14	2.00	−0.01	1.99	0.0084

## Data Availability

Data is contained within the article or Appendix A.

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
