# Peer review of "Molecular Signature of Small Cell Lung Cancer after Treatment Failure: The MCM Complex as Therapeutic Target"

_cancers, 2021, doi:10.3390/cancers13061187_

Round 1
Reviewer 1 Report
Reviewer's comments on the manuscript entitled, Molecular signature of small cell lung cancer after treatment failure: the MCM complex as therapeutic target authored by Seki and co-authors
Mini chromosome maintenance proteins are components of the pre-replicative complex that are required for the initiation of DNA replication. As a part of the CMG complex, they are important for helicase activity needed for DNA replication. It has been known for a long time that MCMs serve as predictive biomarkers for poor prognosis in many different cancers, including MCM7 as a biomarker for SCLC. Some prior studies have also shown that the suppression of MCM chemosensitizes to 5FU and Gemcitabine in pancreatic cancers.
In the current manuscript, the authors examine expression of 3 SCLC patients after treatment failure and identify a cohort of genes that are upregulated by microarray analyses. They focus on MCM complex and find that the loss of MCM in cell lines shows G1 arrest or a G2/M accumulation. Furthermore, loss of MCM gene enhances cisplatin sensitivity of SCLC cells.
While the quality of the data is good and convincing, there is no novelty in the observations. The use of MCMs as a biomarker is known for a long time. The loss of MCM impacting sensitivity to drugs is well documented. This reviewer recommends that this work be submitted as case studies to a more specialized journal.
Author Response
Reviewer #1
Response: We would like to thank you for your important comments on our paper. The point I would like to emphasize in this paper is to provide molecular signature (protein-coding genes) of autopsy specimens of treatment failure of SCLC. As you know, there are almost no studies on gene expression signatures using SCLC clinical specimens. This is because there are very few cases that are clinically indicated for surgery. The data provided this paper is a signature of autopsy specimens (3 cases) of patients who became refractory to treatment. This data is very useful information for lung cancer researchers, especially drug-resistance of this disease.
The following sentences have been added in Introduction and Discussion parts.
(Introduction)
Our mRNA expression signature data is deposited in GEO (Gene Expression Omnibus: GSE 162102) database and is accessible.
There have been few reports of gene expression signature in treatment failure of SCLC clinical specimens. This signature contains molecules involved in treatment resistance and will contribute to the study of SCLC molecular pathogenesis.
(Discussion)
In order to control cancer cells that have acquired treatment resistance, genome analysis research using specimens of patients who have become treatment resistant is indispensable. We have succeeded in creating a molecular signature using clinical specimens of SCLC patients with treatment failure. Although the number of cases is limited, it provides a valuable signature that has been rarely reported so far.
In this study, we created a molecular signature of patients with SCLC who became treatment failure. This signature contains a number of molecules that may control treatment resistance. Based on this signature, the understanding of molecular pathogenesis for treatment-resistant SCLC will be accelerated.
Thank you for your constructive comments and suggestions. We believe that our manuscript has been greatly improved and is now suitable for publication in Cancers. Again, thank you for your consideration of our manuscript for publication in your journal.
Sincerely yours,
Naohiko Seki, Ph.D.
Department of Functional Genomics
Reviewer 2 Report
This paper tried to identify a gene expression signature of SCLC after treatment failure followed by pathway analysis. From the enriched pathway, the authors focused on the cell cycle (which was in the top rank of the statistical significance) and investigated four genes associated with this pathway: MCM 2, 4, 6 and 7. Using siRNA mediated knockdown experiments, the authors demonstrated the functional effect of these genes for cancer cells.
Overall, this paper demonstrates plenty of data, including transcriptome data from SCLC specimens and in vitro functional assays. The logic is consistent and the conclusion is appropriate. Although the roll of MCM family for lung cancer is known in some degree, this paper may build up a new knowledge stone for SCLC.
However, some corrections are recommended as following:
- For microarray expression analysis, the authors compared data from cancer vs normal tissues. Though, cancer tissues included not only lung cancer tissue (primary lesion or metastasis of case no 2 and no 3) but also liver (metastasis of case no. 1) and lymph node (case no. 1 and no. 2). Questions raised for the logic of comparison. Whether tissue specificity and personal differences were considered? Please, explain the logic used for the identification of gene expression signature in the results section. It might be good if the authors describe the limitation of samples (div. cancer tissues, small numbers etc) in discussion sections.
- For Fig 5 (cell cycle regulation) and 6 (apoptosis), the authors used one siRNA for each genes although they used two siRNA for Fig 4 (proliferation). Please clarify, which of those siRNA were used for Fig 5 and 6. (e.g. instead of si-MCM2, the name should be given as si-MCM2-1 or si-MCM2-2)
Author Response
Reviewer #2
Comment-1: For microarray expression analysis, the authors compared data from cancer vs normal tissues. Though, cancer tissues included not only lung cancer tissue (primary lesion or metastasis of case no 2 and no 3) but also liver (metastasis of case no. 1) and lymph node (case no. 1 and no. 2). Questions raised for the logic of comparison. Whether tissue specificity and personal differences were considered? Please, explain the logic used for the identification of gene expression signature in the results section. It might be good if the authors describe the limitation of samples (div. cancer tissues, small numbers etc) in discussion sections.
Response: The reviewer's indication is an important issue in this study. The point I would like to emphasize in this paper is to provide molecular signature (protein-coding genes) of autopsy specimens of treatment failure of SCLC. As you know, there are almost no studies on gene expression signatures using treatment failure of autopsy specimens. The data provided this paper is a signature of autopsy specimens (3 cases) of patients who became refractory to treatment. This data is very useful information for lung cancer researchers, especially drug-resistance of this disease.
As suggested by the reviewer’s comment, the following sentences have been added.
(Discussion)
In order to control cancer cells that have acquired treatment resistance, genome analysis research using specimens of patients who have become treatment resistant is indispensable. We have succeeded in creating a molecular signature using clinical specimens of SCLC patients with treatment failure. Although the number of cases is limited, it provides a valuable signature that has been rarely reported so far.
Comment-2: For Fig 5 (cell cycle regulation) and 6 (apoptosis), the authors used one siRNA for each genes although they used two siRNA for Fig 4 (proliferation). Please clarify, which of those siRNA were used for Fig 5 and 6. (e.g. instead of si-MCM2, the name should be given as si-MCM2-1 or si-MCM2-2)
Response: Thank you for pointing it out. Cell cycle and apoptosis experiments were conducted using siRNAs (two types siRNAs) used in Fig. 4 (cell proliferation assay). I present the revised figures (Fig.5 and Fig.6). With the addition of experiments, sentences (Figure legend) have been added. The added text have highlighted in yellow, please confirm it.
Thank you for your constructive comments and suggestions. We believe that our manuscript has been greatly improved and is now suitable for publication in Cancers. Again, thank you for your consideration of our manuscript for publication in your journal.
Reviewer 3 Report
Both the design and methodology of the study were chosen correctly and prove the presented thesis. The prepared figures are legible and consistent with the description. The text is comprehensible and free of major grammatical or stylistic errors.
However some minor issues occurs and they need to be addressed:
- The manuscript (including references) should be well-checked.
- Were the cells analysed for mycoplasma if so, please indicate the test used.
- Material and method section - The group of patients who died of drug-resistant SCLC can be higher (3 is very poor). I think that this group should be expanded.
- Results sextion - Figure 2A and 2B - please add scale line. Where is the control photo? Please add.
In connection with the above, I recommend the article to be accepted for publication.
Author Response
Reviewer #3
Comment-1: The manuscript (including references) should be well-checked.
Response: I appreciate for your suggestions. I checked the text and references again.
Comment-2: Were the cells analyzed for mycoplasma if so, please indicate the test used.
Response: No confirmation of mycoplasma infection was made during this experiment.
Comment-3: Material and method section - The group of patients who died of drug-resistant SCLC can be higher (3 is very poor). I think that this group should be expanded.
Response: The reviewer's indication is an important issue in this study. Our concept is that analysis of patient specimens that have become drug-resistant is essential to control treatment resistance. Based on this concept, we have spent several years creating this signature for autopsy specimens (3 cases). Molecular signature of SCLC creation will continue in the future.
As suggested by the reviewer’s comment, the following sentences have been added.
(Discussion)
In order to control cancer cells that have acquired treatment resistance, genome analysis research using specimens of patients who have become treatment resistant is indispensable. We have succeeded in creating a molecular signature using clinical specimens of SCLC patients with treatment failure. Although the number of cases is limited, it provides a valuable signature that has been rarely reported so far.
Comment-4: Results section - Figure 2A and 2B - please add scale line. Where is the control photo? Please add.
Response: Following your indication, I presented the scale bar in Figure 2A and 2B.
Thank you for your constructive comments and suggestions. We believe that our manuscript has been greatly improved and is now suitable for publication in Cancers. Again, thank you for your consideration of our manuscript for publication in your journal.
Sincerely yours,
Naohiko Seki, Ph.D.
Department of Functional Genomics
Chiba University Graduate School of Medicine
Round 2
Reviewer 1 Report
This reviewer recommends that this work be submitted to a more specialized journal.
Author Response
Revise letter (cancers-1065029)
Cancers
Special Issue "Lung Cancer: Targeted Therapy and Immunotherapy”
February 26, 2021
Dr. Noriaki Sunaga
Guest Editor, Cancers
Department of Respiratory Medicine,
Gunma University Graduate School of Medicine,
Maebashi, Gunma, Japan
Dear Dr. Sunaga,
We would like to express our gratitude for your consideration of our above-mentioned manuscript for publication in Cancers. Enclosed, please find the re-revised manuscript (cancers-1065029) along with a detailed explanation of the revisions, which were made based on the reviewers’ comments. All changes are highlighted in the revised manuscript.
Reviewer #1
We would like to thank you for taking the time to comment on our submitted paper. We consider this special issue to be a special journal of SCLC. We believe that “Molecular signature of treatment failure patients with SCLC” provides important data for lung cancer researchers. We are spending a lot of effort to get this expression signatures. Moreover, we have a responsibility to pay tribute to the patients and their families who donated the valuable specimens. Please understand this research.
Thank you for your constructive comments and suggestions. We believe that our manuscript has been greatly improved and is now suitable for publication in Cancers. Again, thank you for your consideration of our manuscript for publication in your journal.
Sincerely yours,
Naohiko Seki, Ph.D.
Department of Functional Genomics
Chiba University Graduate School of Medicine
1-8-1 Inohana, Chuo-ku,
Chiba 260-8670, Japan
Phone: +81-43-226-2971
Fax: +81-43-227-3442
e-mail: naoseki@faculty.chiba-u.jp
